# Utility of Diffusion and Magnetization Transfer MRI in Cervical Spondylotic Myelopathy: A Pilot Study

**DOI:** 10.3390/diagnostics12092090

**Published:** 2022-08-29

**Authors:** Hea-Eun Yang, Wan-Tae Kim, Dae-Hyun Kim, Seok-Woo Kim, Woo-Kyoung Yoo

**Affiliations:** 1Department of Rehabilitation Medicine, VHS Medical Center, Seoul 05368, Korea; 2Department of Radiology, VHS Medical Center, Seoul 05368, Korea; 3Department and Research Institute of Rehabilitation Medicine, Yonsei University College of Medicine, Seoul 03722, Korea; 4Department of Orthopaedic Surgery, Hallym University Sacred Heart Hospital, Anyang 14068, Korea; 5Department of Physical Medicine and Rehabilitation, Hallym University Sacred Heart Hospital, Anyang 14068, Korea

**Keywords:** cervical spondylotic myelopathy, MRI, tract-specific analysis

## Abstract

Diffusion tensor imaging (DTI) and magnetization transfer (MT) magnetic resonance imaging (MRI) can help detect spinal cord pathology, and tract-specific analysis of their parameters, such as fractional anisotropy (FA), mean diffusivity, axial diffusivity (AD), radial diffusivity (RD) and MT ratio (MTR), can give microstructural information. We performed the tract-based acquisition of MR parameters of three major motor tracts: the lateral corticospinal (CS), rubrospinal (RuS) tract, and lateral reticulospinal (RS) tract as well as two major sensory tracts, i.e., the fasciculus cuneatus (FC) and spinal lemniscus, to detect pathologic change and find correlations with clinical items. MR parameters were extracted for each tract at three levels: the most compressed lesion level and above and below the lesion. We compared the MR parameters of eight cervical spondylotic myelopathy patients and 12 normal controls and analyzed the correlation between clinical evaluation items and MR parameters in patients. RuS and lateral RS showed worse DTI parameters at the lesion level in patients compared to the controls. Worse DTI parameters in those tracts were correlated with weaker power grasp at the lesion level. FC and lateral CS showed a correlation between higher RD and lower FA and MTR with a weaker lateral pinch below the lesion level.

## 1. Introduction

The incidence of cervical spondylotic myelopathy increases with age. The clinical symptoms are diverse, and researchers have obtained various imaging findings associated with the disease [1]; however, the symptoms and imaging findings do not necessarily match [2,3]. Researchers recommend diffusion tensor imaging (DTI) and magnetization transfer (MT) magnetic resonance (MR) imaging (MRI) for the detection of spinal cord pathology that cannot be identified with conventional MRI.

DTI and MT MRI provide variable parameters that enable quantitative studies. Fractional anisotropy (FA) and mean diffusivity (MD) show good consistency and are the most frequently used DTI parameters. FA reveals both demyelination and axonal injury; however, they are not specific to the underlying cause. MD measures membrane density and is sensitive to cellularity, edema, and necrosis. FA values decrease and MD values increase in injured or diseased spinal cords [4,5,6]. Axial diffusivity (AD) and radial diffusivity (RD) are also used; however, they are less frequently used than FA and MD because of their deceptive change in complex tissue architecture and mutual effect on each other. AD specifically reveals axonal content. AD decreases with acute traumatic spinal cord injury because of axon fragmentation, while it increases over time in chronic neurodegenerative disease because axon fragments are cleared. RD reveals the myelin status relatively specifically and increases with demyelination [7,8,9,10].

MT MRI provides an indirect surrogate for myelin content and is specific to demyelination and degeneration. The MT ratio (MTR) is calculated as the relative change in intensity when the images are acquired with and without an off-resonance MT pulse. MTR in the white matter (WM) is reduced in demyelinating diseases. Reduced MTR values increase as remyelination and gliosis occur but do not reach the values at baseline before demyelination [11,12,13,14,15].

Numerous studies using DTI and MT have been conducted to confirm the pathology of the cervical spinal cord. In one study, FA and apparent diffusion coefficient (ADC) were analyzed at the maximal compression level and above the compression level (C2/C3) in patients with cervical spondylotic myelopathy. FA at the compression level and ADC above and at compression levels demonstrated significant differences between the patients and the controls. Among the patients, the FA value differed based on the presence of symptoms; however, the ADC value was not related to the symptoms [4]. Another study on compressive cervical myelopathy reported lower FA values in patients than in controls at the proximal spinal cord level (C2/C3). The FA values at the compression level were not analyzed in this study [5]. In a study on MT MRI in patients with traumatic cervical spinal cord injury (SCI), MT saturation of the cervical cord at the C2/C3 level was associated with the pin-prick score [12]. In patients with cervical spondylotic myelopathy, the MTR was lower in the patients than in the controls. In addition, MTR values were inversely correlated with the Nurick score, indicating that a lower MTR value is correlated with more severe the symptoms [16]. However, the level and columns or tracts of the spinal cord were not differentiated in the aforementioned studies. Previous studies have principally analyzed the values at the relatively wide proximal cord or mean values by combining several levels, and the areas of the cord were not divided and analyzed as the entire cord. This is because performing and accurate analysis in a narrow spinal cord area is difficult, and it is even more challenging if the cord is compressed. However, the pathology in the upper and lower spinal cord of the compression lesion is different owing to their longitudinal structure. In addition, findings obtained within the cord vary based on the location, e.g., whether it is in the ascending or descending tract [17,18]. Successful spinal cord tractography depends on many factors, such as image acquisition settings, region of interest design, and distortion correction [19,20]. With the development of imaging techniques and the availability of excellent software, it is now possible to discriminate between detailed structures, such as columns within the spinal cord, which enables tract-specific analysis [15,20,21,22].

We aimed to examine the pathologic changes in cervical spondylotic myelopathy using DTI and MT imaging techniques and confirmed the correlation between the MR parameters (FA, MD, AD, RD, and MTR) and clinical measures. Individual tracts were separately analyzed at lesion levels at the compression as well as above and below the lesion. We hypothesized that tract-specific analysis using DTI and MT imaging can reveal subtle pathologic changes in cervical spondylotic myelopathy, especially in patients with mild cervical spondylotic myelopathy without obvious motor or sensory impairment affecting activities of daily living.

## 2. Materials and Methods

We recruited patients with cervical spondylotic myelopathy. Patients who complained of neck pain, upper extremity paresthesia or radicular pain, motor weakness, and spasticity and had confirmed radiographic evidence of cord compression with cervical spondylosis were included. Eight patients were enrolled after excluding patients with previous spinal surgery, cervical cord compression of other than degenerative origin, and peripheral nerve lesions that could cause sensory symptoms. The Institutional Review Board of VHS Medical Center approved all the procedures of the study, and written informed consent was obtained from each participant. The Douleur Neuropathique 4 (DN4) questionnaire was employed to assess neuropathic pain. DN4 consists of seven self-report items and three clinical examination questions and is a valid screening tool for patients with central and peripheral neuropathic pain [23]. Scores range from 0 to 10 points, and a score of ≥4 points is considered positive [24]. The Japanese Orthopaedic Association (JOA) score, the most frequently used scale to evaluate functional status in patients with cervical myelopathy, was also administered. This scale assesses motor dysfunction in the upper and lower extremities; sensory function in the upper extremities, trunk, and lower extremities; and bladder function. Scores range from 0 to 17 points and a score of ≥13 points is considered as mild dysfunction [25]. The presence or absence of gait abnormality, spinal ataxia, pyramidal signs, and hyperreflexia were evaluated. None of the patient presented gait abnormality, spinal ataxia, or pyramidal signs, such as Hoffman sign, Babinski sign, and ankle clonus. Increased bicep or brachioradialis reflexes were observed in some patients. We evaluated the American Spinal Cord Injury Association sensory and motor scores, lateral pinch, and power grasp. The maximum compression level was determined by calculating the compression ratios at each intervertebral level using conventional MR images. T2 signal changes occurred in some participants. The clinical data are summarized in Table 1.

The control group consisted of 12 healthy individuals; however, they were not age-matched. However, as spondylosis is attributed to degeneration, age-matched healthy individuals may also have mild spondylosis [1,26]. Therefore, we selected young individuals as controls. Table 2 outlines the differences between the patients and the controls.

### 2.1. MRI Acquisition

#### 2.1.1. Conventional Imaging

Acquisitions were conducted using a 3-Tesla MRI system (Skyra, Siemens, Germany). T1-weighted images were obtained according to the following parameters: axial orientation, one slab of 192 slices; slice thickness, 1.0 mm; field of view (FOV), 320 mm; time repetition (TR), 2000 ms; echo time (TE), 3.72 ms; voxel size, 1.0 × 1.0 × 1.0 mm^3^; flip angle, 9°; phase oversampling, 0%; slice oversampling, 0%; bandwidth, 150 Hz/pixel; and turbo factor, 192. T2-weighed images were obtained according to the following parameters: axial orientation, one slab of 64 slices; slice thickness, 0.8 mm; FOV, 256 mm; TR, 1500 ms; TE, 119 ms; voxel size, 0.8 × 0.8 × 0.8 mm^3^; flip angle, 120°; phase oversampling, 80%; slice oversampling, 12.5%; bandwidth, 625 Hz/Pixel; and turbo factor, 88.

#### 2.1.2. DTI

DTI was obtained using single-shot echo-planar imaging with the following parameters: b value, 800 s/mm^2^; 64 diffusion gradient directions; slice thickness, 5 mm; FOV, 86 mm; TR, 3300 ms; TE, 93 ms; voxel size, 0.9 × 0.9 × 5.0 mm; and bandwidth, 882 Hz/Pixel.

#### 2.1.3. MT Imaging

We acquired T1-weighted three-dimensional gradient echo images with slab-selective excitation, with and without MT saturation pulses using the following parameters: axial orientation, 17 slices; FOV, 160 mm; TR, 600 ms; TE, 4.83 ms; voxel size, 0.6 × 0.6 × 5.0 mm^3^; flip angle, 25°; phase oversampling, 80%; and bandwidth, 380 Hz/Pix.

Images were processed using open-source software packages, Spinal Cord Toolbox (SCT), and the PAM50 template [27]. The program is available on the website; https://spinalcordtoolbox.com/user_section/installation.html (accessed on 5 August 2021). SCT has been used in cervical cord analysis, even in the presence of pathologies such as traumatic spinal cord injury and myelopathic spinal cord compression, and has proven variability and validity [11,15,17]. DTI parameters, including FA, MD, AD, and RD values, were obtained. We computed the voxel-wise MTR using the following equation: ((S0−SMT)/S0) × 100, where S0 and SMT are T1-weighted images with and without the MT pre-saturation pulse, respectively [15].

MR parameters from three levels were selected as follows: above the lesion level (C2/C3), at the most compressed lesion level, and below the lesion level (C7/T1). The level of the most compressed lesion was determined by calculating the compression ratio [28]. In healthy controls, above the lesion and below the lesion level were analyzed in C2/3 and C7/T1 as in the patient group. As there was no lesion level in the control group, the C5/6 level was selected for comparison with the lesion level of the patients [4,20]. Figure 1 shows the sagittal slices and axial slices of two patients at above the lesion level, the maximum compression level, and below the lesion level. The average of parameters measured from all axial slices contained in each segment were analyzed, and three to four slices were included per segment.

Based on the WM atlas of the human cervical spinal cord, we performed tract-based acquisition of the MR parameters [29]. Among the 15 different WM tracts on each side, three major motor tracts and two major sensory tracts were selected for the analysis. The analyzed motor tracts were as follows: the lateral corticospinal (CS) tract, the rubrospinal (RuS) tract, and the lateral reticulospinal (RS) tract; the sensory tracts were as follows: fasciculus cuneatus (FC) and spinal lemniscus (spinothalamic and spinoreticular tracts) (Figure 2). Alteration of diffusion metrics is significant in the lateral and dorsal columns in the presence of degenerative spinal cord compression [30]. The compressive stress increases in the lateral and dorsal columns when there is acute compression on chronic compressed spinal cord owing to degeneration [31]. The lateral CS tract, RuS tract, lateral RS, spinal lemniscus, and FC tract are located in the lateral and dorsal column, and these are expected to be the most affected.

### 2.2. Data Processing

Template-based analysis was performed using the SCT, a comprehensive and open-source software for analyzing multiparametric MRI data of the spinal cord [27]. The SCT also offers motion correction of MRI data. The analysis consists of six main steps following motion correction: (1) template registration to an anatomic image, (2) template registration to a WM/gray matter (GM) contrasted image, (3) auto segmentation of the WM and GM, (4) template registration of probabilistic WM/GM atlases to the WM/GM segmentations using a multi-label registration method, (5) template registration to a multiparametric MRI parameter image (FA, MD, AD, RD, and MTR), and (6) atlas-based analysis to quantify the parameters in specific sub-regions of the spinal cord [32]. The registration framework is described in Figure 3.

Although automated designation of the spinal cord area was well performed in the non-compressed normal-appearing spinal cord, an error occurred at the deformed spinal cord owing to compression. The spinal cord area was manually masked if there was an error following a thorough review of the automatically selected cord area.

### 2.3. Statistical Analysis

Statistical analysis was performed using the SPSS 19.0 Statistical Package for the Social Sciences (IBM Corp., Armonk, NY, USA) by combining the left and right sides as pooled data [5]. We conducted between-group comparisons using the Mann–Whitney U test, chi–square test, and Fisher’s exact test. Statistical significance was set at *p* < 0.05. Spearman correlation analysis was performed to assess the correlation of each of the three categories of variables (clinical scales, clinical measures, and MR parameters) in patients with cervical spondylotic myelopathy. The false discovery rate was used to perform corrections for multiple comparisons and to better interpret the results.

## 3. Results

### 3.1. Group Comparison

FA was lower and MD, AD, and RD were higher in the patient group in the lateral RS tract at each level. Similar changes in the DTI parameters were noted in the RuS tract and spinal lemniscus at the lesion level. Lower FA and higher MD and RD were observed in the RuS tract, and lower FA was observed in the spinal lemniscus. FC also showed group differences in the DTI parametric values; however, the differences were contrary to expectations. FC showed lower MD, AD, and RD and so-called better DTI parametric values in the patient group above the lesion. The MTR was higher in the patient group, contrary to expectations, in the lateral CS tract, RuS tract, lateral RS tract, and spinal lemniscus below the lesion level. Table 3 summarizes the parameters that were different in each tract at each level.

### 3.2. Correlations between the MR Parameters and Clinical Evaluation Items

All the parameters showing correlations with the clinical evaluation items are summarized in Table 4. In one case, an inverse correlation was observed: the higher the MTR, the worse the symptoms. As the distribution of the correlation seems sporadic, they were grouped as follows and are described in detail: (1) tracts in which the DTI parameters or MTR showed differences between the groups, and correlations with clinical items were also confirmed. (2) DTI parameters and MTR correlated with the clinical items simultaneously.

The RuS tract, lateral RS tract, and spinal lemniscus, which showed group differences in the DTI parameters at the lesion level, also revealed a correlation with the power grasp at the lesion level. A group difference is expressed as a bar graph, and the correlation is presented as a scatter plot in Figure 4. The scatter plots contain 16 data points because the data of the left and right sides were pooled together, as aforementioned. The parameters showing the group differences and the parameters showing the correlation did not necessarily match.

Below the lesion level, the lateral CS tract showed a correlation of the lateral pinch with the DTI parameters and MTR simultaneously. The correlation is presented as a scatter plot in Figure 5. Among the DTI parameters, FA and RD were associated with the MTR.

## 4. Discussion

### 4.1. Group Comparison

It was hypothesized that symptoms of cervical spondylotic myelopathy result from pathologic changes in the specific tract rather than from the compression itself [33]. We intended to detect the changes in the spinal cord tracts in patients exposed to chronic compression due to spondylosis compared to healthy controls. To adjust for the age difference between the groups, a regression analysis with age as a covariate was conducted but was not appropriate because of multicollinearity. To complement this, regression analysis was conducted to control for within-group age differences. Tracts showing expected results were the lateral RS tract at all levels, the RuS tract, and spinal lemniscus at the lesion level. A slow neuroinflammatory response of the spinal cord under chronic compression has been reported in previous animal and human studies [34,35,36]. This slow neuroinflammatory process results in chronic demyelination and causes activation of the microglia and the release of toxic stimuli from cells, such as astrocytes and neurons [33]. Our study demonstrated consistent results through changes in the DTI values of the RuS tract, lateral RS tract, and spinal lemniscus. In particular, the lateral RS tract showed consistent results at all levels, with the most pronounced group difference. Unlike the lateral RS tract, the RuS tract and spinal lemniscus showed differences only at the lesion level, which could occur because the lesion level was most directly affected by compression. The majority of the literature on the function of specific tracts was performed in primates rather than in humans. The RS tract plays an important role in the coordination of activity primarily performed by the corticospinal tract, such as reaching movements in the upper extremities and possibly finger movements, and has been shown to be associated with locomotion and postural adjustment [12,37,38]. The RuS tract is directly connected with motor neurons of the distal upper limb muscles and is associated with dynamic movement control in primates [39]. Both the RS and RuS tracts are also known to play an important role in recovery following corticospinal cord lesions in primates and humans, and increased activity of these tracts is a favorable prognostic factor for restoration of hand function [40,41,42]. In our study, lateral CS did not show any difference in DTI parameters between groups; therefore, the RuS and RS tracts did not show a compensatory increase in activity either. This difference is thought to be attributed to the fact that the patients in our study had relatively milder pathology and symptoms compared to previous studies performed with primates with induced SCI and human SCI showing clearly impaired hand function. None of the patients in our study had significant motor weakness or limitations in independent daily activities. The spinal lemniscus refers to the anterior and lateral spinothalamictracts. The spinothalamic tract is a major pathway of pain transmission, and neuropathic pain is worse if the function of the spinothalamic tract decreases in SCI [43]. Degeneration of the spinothalamic tract was confirmed by MRI, and the degree of degeneration correlated with the degree of impairment in traumatic SCI [44]. DTI parameters reflect the pathology of the spinothalamic tract. Interestingly, Hatem et al. reported that the FA value of the anterior cord area, where the spinothalamic tract is located, correlated better with somatosensory dysfunction than the FA value of the whole cord or posterior column [45]. Our study also showed lower FA in the spinal lemniscus in patients, which is consistent with the results of previous studies. Considering the DTI parameters, the RuS and lateral RS tracts, which are motor tracts, showed differences in most DTI parameters, whereas spinal lemniscus, which is the sensory tract, only showed differences in the FA values. Our finding confirms that FA is the most sensitive parameter among the DTI parameters. Judging from their proximity to each other, another possible explanation is that the motor tract is more vulnerable to pressure than the sensory tract.

FC in the dorsal column showed group differences in DTI parameters above the level of the lesion. Animal studies have demonstrated that the damaged tract differs based on the position, which is above and below the compression lesion. A previous study reported changes above and below the compression level with DTI and histopathology by artificially inducing SCI with the application of compression to the spinal cord in dogs. The study pathologically confirmed spinal cord degeneration in the dorsal funiculus above the lesion and in the ventral funiculus below the lesion [18]. In a study on cats, changes in FA, generalized fractional anisotropy (GFA), and RD were predominant; however, they were also detected rostral to the injury on the dorsal aspect and caudal to the injury on the ventrolateral aspect [46]. Degeneration does not necessarily occur in the anterograde direction only, and retrograde degeneration also occurs; however, antegrade Wallerian degeneration could explain the location of the pathologic change based on the positional relation with the lesion [15]. FC is an ascending tract that transmits proprioceptive and cutaneous feedback signals to the upper extremities; therefore, the group difference above the level of the lesion, which was confirmed in our study, was consistent with previous studies [47,48]. However, we were unable to explain why the DTI values of some tracts were better in patients than in the controls because this phenomenon has not been previously reported. The mild symptoms of the patients enrolled in this study compared to other studies could have contributed to this result. The possibility of repair or regeneration of the spinal cord tracts was assessed; however, it seems unlikely. A literature review on the mechanisms of central nervous system axonal regeneration and remyelination by Uyeda et al. identified that astrocytes, oligodendrocytes, and glial cells play critical roles in the repair process [49]. With degeneration, myelin content decreases and astrocytes become hypertrophied, and these cellular changes are reflected in the DTI parameters as a decrease in FA and an increase in MD, AD, and RD. With regeneration, the myelin content and hypertrophied astrocytes recover, and these changes are reflected in the DTI parameters as an increase in FA, as well as a decrease in MD, AD, and RD. However, recovery of the DTI parameter values did not reach or exceed the baseline level [50,51]. Better DTI values in the patient group in our study cannot be explained as regeneration because they cannot exceed the baseline value, even when regeneration occurs. A compensation mechanism may exist in FC; however, it is difficult to explain this unexpected finding because the number of patients was small and the patients had only mild symptoms.

The MTR, lateral CS tract, RuS tract, lateral RS tract, and spinal lemniscus tracts showed group differences in the MTR below the lesion level, and a higher MTR was observed in the patient group, contrary to expectations. Our findings are not consistent with those of Cloney et al., who reported that the MTR of the anterior cord region was associated with the clinical outcomes [52]. A post-mortem study of humans demonstrated that the MTR value has a strong correlation with the myelin content [53]. MTR indicates demyelination and reduction sensitively, even before demyelination becomes visible on T2-weighted images and increases again with remyelination; however, the value does not reach the baseline, similar to the DTI parameters [54]. One possible explanation for this is that, although myelin layers were formed, the lipid and/or protein content of the newly repaired myelin is biochemically different from mature, compact myelin, affecting the magnetization transfer ratio [55]. Therefore, it is difficult to determine the extent of remyelination using only the MTR values [56]. Moreover, despite its sensitivity to the tissue content, the MTR is not specific to the underlying pathology that is affected by free water MR parameters, which are modulated by other factors, such as inflammation [57]. The level or the location of the tracts with higher MTR in our study is inconsistent, and considering the characteristics of MTR, it is difficult to ascertain the significance of this result.

As the control group was not age-matched, it was necessary to clarify the effect of aging. Previous studies have demonstrated that FA and MTR increase with aging; in contrast, MD and RD increases with aging. The effect of age on AD value is not consistent across reports [58,59,60]. Age itself may have influenced the changes in MR metrics in the patient group. However, as the differences in MR in our study were correlated with the lesion level, it is clear that this change was attributed to spondylotic compression.

### 4.2. Correlations between the MR Parameters and Clinical Items

Although the analysis was performed using advanced analysis technology, it is difficult to ascertain the significance of every single result demonstrating the correlation with the MR parameters. Moreover, previous studies have claimed that caution is needed in the interpretation of the MR parameters. Several physical parameters can influence the DTI parameters, including myelination, axonal density, axonal diameter, and orientation of the fiber bundles [61,62]. RD and AD can cause a deceptive change in the voxels characterized by crossing fibers [9]. The direction of diffusivity is not always preserved in pathological tissue and is not always aligned with the underlying expected tissue architecture [63]. It is necessary to pay attention to the interpretation of the DTI parameters; however, it can be considered a significant finding when there is meaningful consistency. Therefore, in our study, the following cases were judged to be significant, and the results were examined in detail: (1) the tracts in which group differences were confirmed and correlations with clinical items were also confirmed within the patient groups; (2) the tracts in which the DTI parameter and the MTR showed a correlation in the same direction as the clinical items, although there was no group difference. Evidence exists that combining DTI and MT is more specific and reproducible. Reich et al. reported that combining measures of MT and diffusion-weighted MRI is a means to becoming more specific to WM pathology [64]. Smith et al. also reported that MT and diffusion-weighted MRI together provide high reproducibility in the human cervical cord at 3T as well as a robust assessment of the WM pathology [65]. Cohen-Adad et al. identified that combining high angular diffusion imaging with MT is a promising approach to gaining specificity in characterizing spinal cord pathways in traumatic injury [15]. Figure 6 presents the schematic interpretation process of this study concerning the MR parameters.

The lateral RS tract, RuS tract, and spinal lemniscus, which showed worse DTI parameters in patients than in controls at the lesion level, showed a correlation of the DTI parameters with the power grasp at the lesion level. It is known that the RS and RuS tracts are involved in hand motor function. The RS tract contributes to hand motor tasks involving gross finger manipulations, but to a lesser extent, to more dexterous finger movements in humans with cervical SCI [32]. Recovery of gross hand movement is impaired when RuS tract involvement is present [34]. Our results corroborate previous findings that the lateral RS and RuS tracts contribute to gross hand function. The DTI parameters of the lateral RS and RuS tract were well correlated with the power grasp, explaining the patient’s subtle symptoms well. It is difficult to specify tract-specific pathology via conventional MRI; therefore, tract-specific analysis provides a good chance to identify the pathology in detail and understand the associated symptoms. Spinal lemniscus was associated with lower FA values in patients than controls and FA values were correlated with power grasp at the lesion level. There are also some differences in the results of the spinal lemniscus compared with the lateral RS and RuS tracts. In the group comparison, there were differences in three or more DTI parameters in the lateral RS and RuS tracts; however, only one DTI parameter, FA, was different in the spinal lemniscus. While the parameters showing group differences and those showing correlation overlapped in the lateral RS and RuS tracts, parameters showing group differences and parameters showing correlation did not match in the spinal lemniscus. Even if correlation is confirmed in the tracts showing differences between the groups, it is difficult to determine the unity between the DTI parameters.

The DTI parameters and the MTR showed a correlation in the same direction with clinical features in the lateral CS tract. DTI parameters and MTR showed a positive correlation with lateral pinch below the lesion level in the lateral CS tract. Among the DTI parameters, FA showed correlations; thus, we can see that FA is closely associated with MTR. FA is the most commonly used DTI parameter because it sensitively measures microstructural integrity, although it is not specific to the underlying pathology. In our study, FA showed the most significant DTI results, confirming that FA is a sensitive parameter. MTR increases when there are more proteins and lipids in the myelinated sheath [57]. This is the only significant finding that was demonstrated in the lateral CS tract, the major motor pathway, in our study. There was no difference between the groups in the lateral CS tract, and we assumed that this was attributed to the fact that the patients included in our study only had mild symptoms. However, as the MTR values of the lateral CS tract below the lesion level correlated with the lateral pinch, a change in the myelin state of the lateral CS tract was observed in the patients. 

One of the greatest challenge when acquiring MR images in the spinal cord is the inhomogeneous magnetic field in the region [65,66,67], and there are certain possible errors in DTI-based tractography due to gradient inhomogeneity in the spatial context-B matrix spatial distribution. The SCT we used in our study provides few correction techniques. Thus, we applied the optimal MRI acquisition protocol for the processing and analysis using SCT. Correction was performed using slice-wise translation in the axial plane, which in the SCT can be performed using regularization along the z direction. In addition to this feature, correction in SCT includes the methods of Xuet et al. for grouping successive volumes, thereby improving the robustness of registration in high b-value diffusion MRI data. This approach is particularly successful for correcting slow drifts in long acquisitions [67]. Nevertheless, the possibility that diffusion tensor values were affected by using a single device cannot be completely excluded. Even if the average eigenvalue shifted, the difference between the patient and the control groups can be confirmed without looking at the absolute value [68].

The small sample size and lack of age-matched controls are certain limitations of this study. To increase the reliability of our results, future studies with a larger number of patients with age-matched controls should be conducted.

## 5. Conclusions

Through tract-specific analysis, the difference between the control group and patients, and the correlation between individual tracts and clinical symptoms, could be identified in cervical spondylotic myelopathy. Tract-specific analysis and interpretation of MR parameters should be considered from various perspectives. If the DTI parameter and MTR show unity, it would be reliable, and among the DTI parameters, FA is the most related to MTR. If only the DTI parameters are reviewed without considering the MTR, uniformity should exist among the DTI parameters. The positional relationship between the tracts within an axial plane must be considered when determining the MR parameters of each tract. Considering all the aforementioned points, we obtained the following: RuS and lateral RS showed worse DTI parameters at the lesion level in patients than in the controls, and worse DTI parameters in those tracts were correlated with weaker power grasp at the lesion level. Lateral CS showed a correlation between higher RD and lower FA and MTR, with a weaker lateral pinch below the lesion level.

## Figures and Tables

**Figure 1 diagnostics-12-02090-f001:**
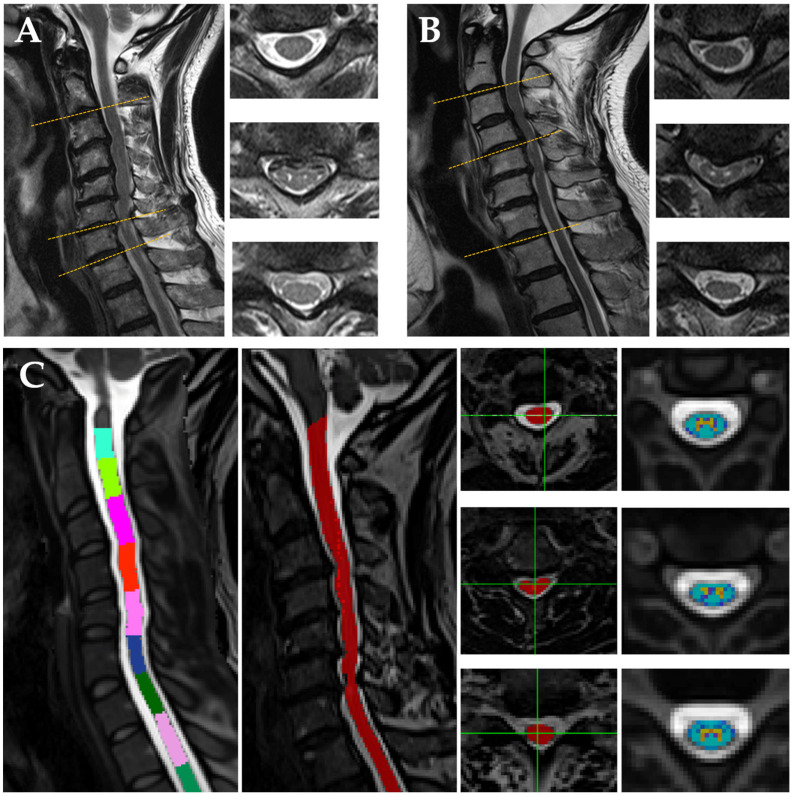
Examples of sagittal and axial slices at each level in two patients. The maximum compression level is at the C6/7 (**A**) and at the C4/5 (**B**). Panel (**C**) shows vertebral labeling, cord area at each level, and white matter/gray matter segmentation of patient A. More details concerning data processing are presented below.

**Figure 2 diagnostics-12-02090-f002:**
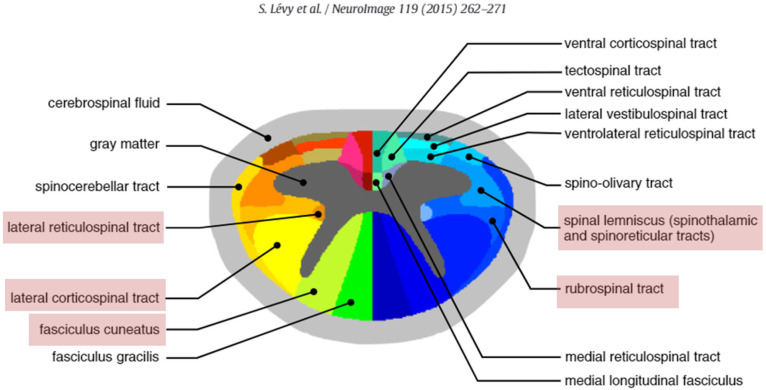
Spinal cord white matter tracts. The colored atlas of the spinal cord white matter has been adapted from Lévy et al., 2015 [29].

**Figure 3 diagnostics-12-02090-f003:**
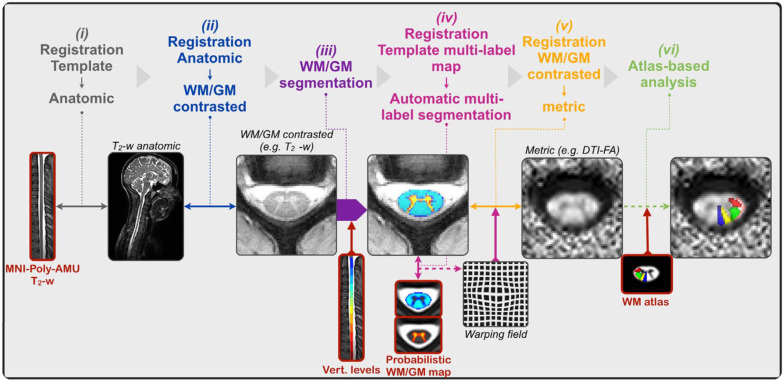
Registration framework for the extraction of metrics using the template. Template-based elements are shown in red. The figure has been adapted from Dupont et al., 2017 [32].

**Figure 4 diagnostics-12-02090-f004:**
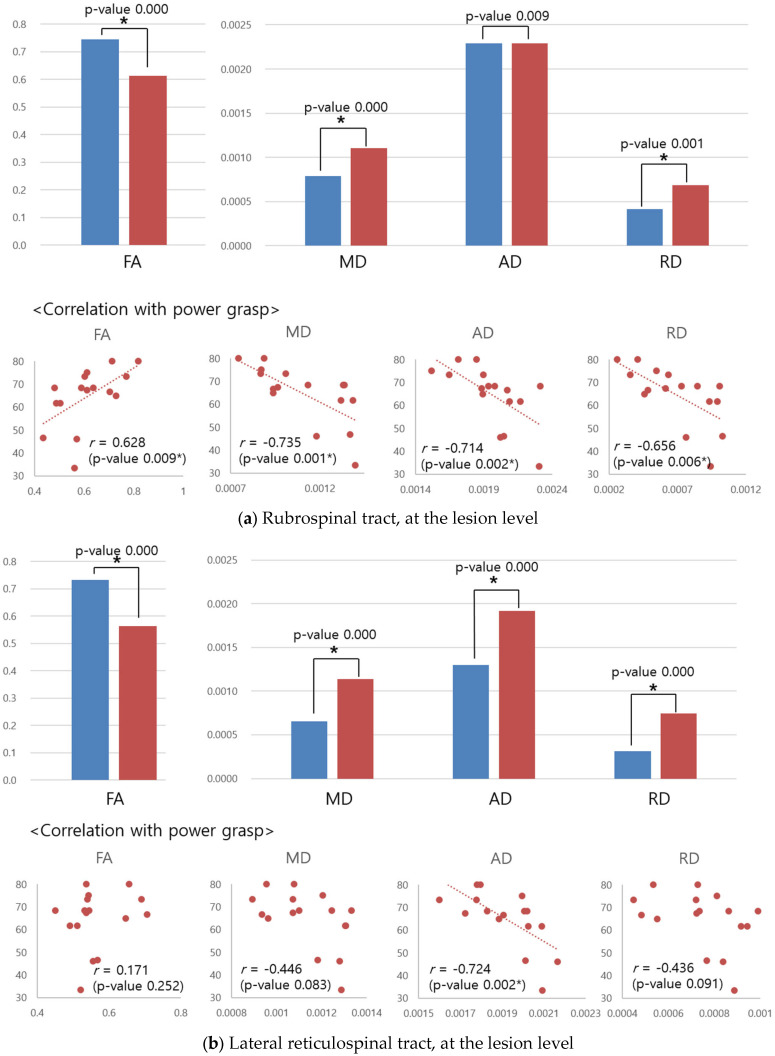
Group difference and correlation between MR parameters and motor items at rubrospinal tract (**a**), lateral reticulospinal tract (**b**), and spinal lemniscus (**c**). * statistically significant.

**Figure 5 diagnostics-12-02090-f005:**
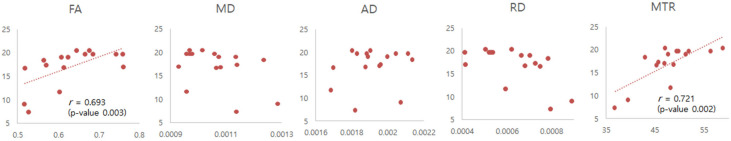
Correlation between MR parameters and lateral pinch at lateral corticospinal tract.

**Figure 6 diagnostics-12-02090-f006:**
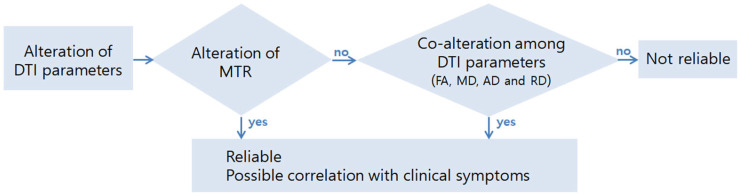
Schematic interpretation process of MR parameters.

**Table 1 diagnostics-12-02090-t001:** Clinical information of the patients.

Patient No.	1	2	3	4	5	6	7	8
Level of maximal compression	C3/4	C5/6	C4/5	C4/5	C6/7	C5/6	C5/6	C6/7
Compression ratio at level of maximal compression	0.30	0.33	0.39	0.40	0.49	0.44	0.61	0.44
T2 high signal intensity	+	+	+	-	-	-	-	-
DN4 score	2	2	8	4	4	3	5	4
JOA score	16	15	13	15	16	16	16	16
Hyperreflexia	-	+	-	-	+	+	-	-
Lateral pinch (Rt./Lt.)	19.0/19.6	19.0/19.6	9.0/7.3	20.3/11.6	16.6/16.8	18.3/17.3	19.6/17.0	20.3/19.6
Power grasp (Rt./Lt.)	68.3/61.6	68.3/61.1	46.0/33.3	75.0/46.6	73.3/67.3	68.3/80.0	65.0/66.6	80.0/73.3
ASIA motor (Rt./Lt.)	50/50	50/50	38/37	45/44	50/50	50/50	49/49	49/49
ASIA sensory (Rt./Lt.)	111/112	112/112	117/117	112/112	111/110	112/112	108/108	110/110
ASIA light touch (Rt./Lt.)	55/56	56/56	58/58	56/56	55/55	56/56	54/54	55/55
ASIA pinprick (Rt./Lt.)	56/56	56/56	59/59	56/56	56/55	56/56	54/54	55/55

DN4, Douleur Neuropathique 4; JOA, Japanese Orthopaedic Association; ASIA, American Spinal Cord Injury Association.

**Table 2 diagnostics-12-02090-t002:** Demographic information of each group.

	Patient (*n* = 8)	Control (*n* = 12)	*p*-Value
Sex (male:female)	8:0	8:4	0.238
Age (age ± SD)	73.00 ± 5.78	37.08 ± 5.38	0.000 *

SD, standard deviation; * statistically significant.

**Table 3 diagnostics-12-02090-t003:** Group differences of MR parameters of spinal cord tracts at each level.

		Above the Lesion Level			At the Lesion Level			Below the Lesion Level		
		Control	Patient		Control	Patient		Control	Patient	
		Mean ± SD	Mean ± SD	*p*	Mean ± SD	Mean ± SD	*p*	Mean ± SD	Mean ± SD	*p*
Lat CS	FA	0.689 ± 0.083	0.663 ± 0.071	0.464	0.648 ± 0.077	0.597 ± 0.114	0.217	0.602 ± 0.073	0.631 ± 0.081	0.411
	MD	1101 × 10^−6^ ± 0.000	972 × 10^−6^ ± 0.000	0.199	1074 × 10^−6^ ± 0.000	1125 × 10^−6^ ± 0.000	0.746	1240 × 10^−6^ ± 0.001	0.001 ± 1061 × 10^−6^	0.232
	AD	1955 × 10^−6^ ± 0.000	1788 × 10^−6^ ± 0.000	0.256	1849 × 10^−6^ ± 0.000	1950 × 10^−6^ ± 0.000	0.533	2039 × 10^−6^ ± 0.001	0.002 ± 1917 × 10^−6^	0.434
	RD	641 × 10^−6^ ± 0.000	564 × 10^−6^ ± 0.000	0.367	626 × 10^−6^ ± 0.000	712 × 10^−6^ ± 0.000	0.382	756 × 10^−6^ ± 0.000	0.001 ± 633 × 10^−6^	0.146
	MTR	44.707 ± 3.955	46.966 ± 3.196	0.156	43.300 ± 3.893	46.393 ± 4.033	* 0.060 *	42.424 ± 4.481	47.884 ± 5.504	0.006 *
RuS	FA	0.748 ± 0.089	0.718 ± 0.088	0.454	0.745 ± 0.100	0.614 ± 0.109	0.002 *	0.655 ± 0.093	0.688 ± 0.088	0.436
	MD	859 × 10^−6^ ± 0.000	905 × 10^−6^ ± 0.000	0.635	787 × 10^−6^ ± 0.000	114 × 10^−6^ ± 0.000	0.002 *	979 × 10^−6^ ± 0.000	912 × 10^−6^ ± 0.000	0.492
	AD	1763 × 10^−6^ ± 0.000	1796 × 10^−6^ ± 0.000	0.866	1670 × 10^−6^ ± 0.000	1957 × 10^−6^ ± 0.000	0.300	1945 × 10^−6^ ± 0.000	1795 × 10^−6^ ± 0.000	0.419
	RD	425 × 10^−6^ ± 0.000	459 × 10^−6^ ± 0.000	0.638	416 × 10^−6^ ± 0.000	687 × 10^−6^ ± 0.000	0.005 *	653 × 10^−6^ ± 0.000	467 × 10^−6^ ± 0.000	*0.082*
	MTR	47.318 ± 4.182	48.942 ± 3.389	0.367	45.676 ± 5.033	45.877 ± 4.156	0.959	44.450 ± 3.608	50.966 ± 5.324	0.004 *
Lat RS	FA	0.796 ± 0.137	0.646 ± 0.082	0.001 *	0.733 ± 0.124	0.564 ± 0.072	0.001 *	0.781 ± 0.086	0.614 ± 0.057	0.001 *
	MD	646 × 10^−6^ ± 0.000	928 × 10^−6^ ± 0.000	0.000 *	652 × 10^−6^ ± 0.000	1138 × 10^−6^ ± 0.000	0.000 *	779 × 10^−6^ ± 0.000	1030 × 10^−6^ ± 0.000	0.015 *
	AD	1327 × 10^−6^ ± 0.000	1704 × 10^−6^ ± 0.000	0.002 *	1297 × 10^−6^ ± 0.000	1919 × 10^−6^ ± 0.000	0.002 *	1550 × 10^−6^ ± 0.000	1844 × 10^−6^ ± 0.000	0.006 *
	RD	254 × 10^−6^ ± 0.000	540 × 10^−6^ ± 0.000	0.000 *	311 × 10^−6^ ± 0.000	746 × 10^−6^ ± 0.000	0.000 *	295 × 10^−6^ ± 0.000	623 × 10^−6^ ± 0.000	0.000 *
	MTR	46.813 ± 3.449	47.413 ± 3.810	0.724	46.309 ± 2.927	46.376 ± 3.712	0.989	44.370 ± 4.274	48.131 ± 5.034	0.048 *
FC	FA	0.675 ± 0.070	0.676 ± 0.079	0.989	0.597 ± 0.068	0.616 ± 0.101	0.635	0.603 ± 0.060	0.602 ± 0.069	0.989
	MD	1166 × 10^−6^ ± 0.000	920 × 10^−6^ ± 0.000	0.006 *	1124 × 10^−6^ ± 0.000	1079 × 10^−6^ ± 0.000	0.635	1164 × 10^−6^ ± 0.000	1101 × 10^−6^ ± 0.000	0.532
	AD	2189 × 10^−6^ ± 0.000	1753 × 10^−6^ ± 0.000	0.014 *	2033 × 10^−6^ ± 0.000	1934 × 10^−6^ ± 0.000	0.569	2022 × 10^−6^ ± 0.000	1937 × 10^−6^ ± 0.000	0.618
	RD	680 × 10^−6^ ± 0.000	503 × 10^−6^ ± 0.000	0.006 *	759 × 10^−6^ ± 0.000	651 × 10^−6^ ± 0.000	0.111	767 × 10^−6^ ± 0.000	683 × 10^−6^ ± 0.000	0.256
	MTR	44.733 ± 2.768	46.912 ± 4.392	0.207	43.319 ± 2.851	45.640 ± 3.774	* 0.088 *	43.272 ± 4.441	45.287 ± 3.231	0.256
SpLm	FA	0.692 ± 0.100	0.654 ± 0.076	0.367	0.695 ± 0.109	0.595 ± 0.067	0.009 *	0.642 ± 0.086	0.658 ± 0.075	0.644
	MD	995 × 10^−6^ ± 0.000	959 × 10^−6^ ± 0.000	0.746	1043 × 10^−6^ ± 0.001	1120 × 10^−6^ ± 0.000	0.638	1070 × 10^−6^ ± 0.000	1047 × 10^−6^ ± 0.000	0.927
	AD	1718 × 10^−6^ ± 0.000	1722 × 10^−6^ ± 0.000	0.989	1853 × 10^−6^ ± 0.001	1860 × 10^−6^ ± 0.000	0.942	1862 × 10^−6^ ± 0.000	1892 × 10^−6^ ± 0.000	0.910
	RD	533 × 10^−6^ ± 0.000	577 × 10^−6^ ± 0.000	0.638	559 × 10^−6^ ± 0.000	728 × 10^−6^ ± 0.000	0.232	627 × 10^−6^ ± 0.000	625 × 10^−6^ ± 0.000	0.989
	MTR	44.048 ± 5.532	47.904 ± 4.500	* 0.074 *	43.300 ± 3.528	44.623 ± 3.545	0.429	43.168 ± 4.527	49.285 ± 3.990	0.001 *

Lat CS, lateral corticospinal tract; RuS, rubrospinal tract; Lat RS, lateral reticulospinal tract; FC, fasciculus cuneatus; SpLm, spinal lemniscus. Underlined values denote results that are contrary to expectation; * statistically significant.

**Table 4 diagnostics-12-02090-t004:** MR metrics displaying correlation with clinical items. Underline denotes an inverse correlation.

		Lat CS	RuS	Lat RS	FC	Spinal Lemni
Power grasp	above				AD	
	at		FA, MD, AD, RD	AD		MD, AD
	below					
Lateral pinch	above					MTR
	at					
	below	FA, MTR				

Lat CS, lateral corticospinal tract; RuS, rubrospinal tract; Lat RS, lateral reticulospinal tract; FC, fasciculus cuneatus; Spinal Lemni, spinal lemniscus. FA and MTR: positive correlation was considered as normal correlation and negative correlation was considered as inverse correlation; MD, AD, and RD: negative correlation was considered as normal correlation and positive correlation was considered as inverse correlation.

## Data Availability

Data available on request to restrictions. The data presented in this study are available on request from the corresponding author. The data are not pubulicaly available due to participant’s confidentiality.

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
