# Peer review of "Utility of Diffusion and Magnetization Transfer MRI in Cervical Spondylotic Myelopathy: A Pilot Study"

_diagnostics, 2022, doi:10.3390/diagnostics12092090_

Round 1

Reviewer 1 Report

This is a pilot study to evaluate the parameters measured by diffusion tensor imaging (DTI) and magnetization transfer (MT) MRI in 8 patients with mild cervical spondylotic myelopathy compared to twelve young normal controls. Five quantitative parameters, including fractional anisotropy (FA), mean diffusivity (MD), axial diffusivity (AD), radial diffusivity (RD), and MT ratio (MTR) were analyzed from 3 levels in each patient: above the lesion level (C2/C3), at the most compressed lesion level, and below the lesion level (C7/T1). The results showed a trend of worse DTI parameters in patients compared to controls, and also that the parameters were correlated with functional status, such as with weaker power grasp. The methods based on the segmentation of gray matter and white matter in the spinal cord, and the identification of specific white matter tracts using template-mapping, are very interesting. Although the significance of the results was limited by the very small sample size, the described methods may be applied to future studies. Some comments:

1.     Please confirm that only one level at C5/6 was analyzed for the control subjects. If so, the comparison with the results analyzed from other levels in patients may be a major confounding factor that led to unexpected results. Adding an analysis to report the 5 parameters at different levels of the control subjects (C2/3, 3/4, 4/5, 5/6, 6/7, C7/T1) will provide very helpful information to understand the level-dependence results. If a high variation is noted, this might explain why some MTR was higher in patients than in controls.

2.     Please explain how many slices were analyzed. Was the reported parameter from only one 5 mm slice at the center of the level, or the average of parameters measured from all axial slices contained in each color segment on the sagittal view, as shown in Figure 1?

3.     In Table 3, for the diffusivity parameters (MD, AD, RD), please use x10-6 to show results. 0.000 or 0.001 +/- 0.000 does not give sufficient information for readers to see the difference.

4.     Given the lack of strongly significant results, the conclusion is very important for readers to have a good take-home message. It was described that “The positional relationship between the tracts must be considered when determining the MR parameters of each tract.” Adding results from issue 1) will generate more concrete data for understanding the level-dependence variations.

5.     “If the DTI parameter and MTR show unity, it would be reliable, and among the DTI parameters, FA is the most related to MTR. If only the DTI parameters are reviewed without MTR, uniformity should exist among the DTI parameters.” Please explain what does “show unity” and “uniformity” mean. Different parameters may be sensitive to different aspects of the pathology.

Author Response

Point-by-point response to the reviewer's comments is uploaded as a word file.

Reviewer 2 Report

Potentially valuable manuscript.

Major note

1.       DTI results as well as parameters are sensitive to systematic errors. Here we have 64 directions, which can partially eliminate these errors. But due to the fact that we only have one scanner, it is necessary to discuss the possible influence of systematic errors on the parameters of the diffusion tensor. Here the BSD-DTI issue seems to clarify this issue well. Generalized Stejskal Tanner equation for spatially nonuniform gradient fields additionally shows the consequences of systematic errors. These issues need to be carefully considered and their potential impact on the results must be assessed.

2.       The reliability of the results obtained (also in the context of possible systematic errors) should be better discussed. The correlations shown usually have a small R, why not R2? Need to be clearer what is a reliable result, and what maybe just a random result ...

Minor notes

1.       The interpretation of the MT and DTI MRI results should be presented more clearly. Perhaps some form of diagram would be useful. Now it is possible to load, but it is a bit dimly describe

2.        “The T1 parameters were as follows: axial orientation, “

 Here it should be written similarly as below that we have T1 relaxation time weighted images, ...

and for T2 similarly ...

Author Response

(The authors gave the same response as above.)

Round 2

Reviewer 2 Report

Generally, the article is almost ready for publication. I would point out two issues:

- linguistic correction for the clarity of formulated sentences.

- in the discussion on the influence of systematic errors, I would add a sentence about possible errors in DTI-based tractography due to gradient inhomogeneity (in the spatial context - BSD).

Good luck with your further research.

Author Response

Response to reviewer is uploaded as a Word file.
